# Bedrock geochemistry influences vegetation growth by regulating the regolith water holding capacity

Zihan Jiang[1], Hongyan Liu [1✉], Hongya Wang[1], Jian Peng[1], Jeroen Meersmans[2,6], Sophie M. Green[3], Timothy A. Quine [3], Xiuchen Wu[4] & Zhaoliang Song[5]

Although low vegetation productivity has been observed in karst regions, whether and how bedrock geochemistry contributes to the low karstic vegetation productivity remain unclear. In this study, we address this knowledge gap by exploring the importance of bedrock geochemistry on vegetation productivity based on a critical zone investigation across a typical karst region in Southwest China. We show silicon and calcium concentrations in bedrock are strongly correlated with the regolith water loss rate (RWLR), while RWLR can predict vegetation productivity more effectively than previous models. Furthermore, the analysis based on 12 selected karst regions worldwide further suggest that lithological regulation has the potential to obscure and distort the influence of climate change. Our study implies that bedrock geochemistry could exert effects on vegetation growth in karst regions and highlights that the critical role of bedrock geochemistry for the karst region should not be ignored in the earth system model.

[1] College of Urban and Environmental Sciences and MOE Laboratory for Earth Surface Processes, Peking University, 100871 Beijing, China. [2] Cranfield Soil and Agrifood Institute, School of Water, Energy and Environment, Cranfield University, Cranfield MK43 0AL, UK. [3] Geography, College of Life and Environmental Sciences, University of Exeter, Exeter EX4 4SB, UK. [4] Faculty of Geographical Science, Beijing Normal University, 100875 Beijing, China. [5] Institute of Surface-Earth System Science, Tianjin University, 300072 Tianjin, China. [6] Present address: TERRA Teaching and Research Centre, Gembloux Agro-Bio Tech, University of Liège, 5030 Gembloux, Belgium. ✉email: lhy@urban.pku.edu.cn

Understanding the drivers of the spatial variations in vegetation productivity is critical for interpreting and predicting the stability and resilience of terrestrial ecosystems[1–3]. Vegetation productivity is controlled by interactive processes within the atmosphere, pedosphere, and lithosphere that are driven by energy obtained from the sun, $CO_2$ captured from the atmosphere, and water and nutrients absorbed from the regolith[4,5]. However, until recently, most of the studies related to vegetation productivity have been largely confined to climate and topsoil features[6], whereas the importance of deeper belowground components remains poorly understood.

As the lower boundary of Earth's critical zone, bedrock has great potential to influence overlying plants by regulating the chemical and physical properties of regolith[7]. Bedrock is the source of most mineral nutrients (e.g., Fe, P), which subsequently shapes plant growth and community composition[8,9]. Conversely, bedrock also supplies heavy metals (e.g., Hg, Pb and Cd), which can inhibit plant growth[10]. In addition, bedrock can influence the regolith texture and consequently control the water and nutrient retention capacity of the regolith[11]. Recent studies that considered deep regolith and rock samples provided evidence that the chemical composition of bedrock can impart a substantial influence on soil erosion processes, thereby influencing the amounts of water and nutrients retained by the regolith[12,13]. Furthermore, bedrock composition can also be closely linked to ecosystem net carbon gains and losses[14]. However, until now, the lithological controls on plant growth were intuitively considered of secondary importance compared to the effects of climatic and pedological factors[15,16].

In carbonate rock regions, the bedrock compositions strongly influence regolith properties[17] that, in turn, might play the primary role in plant growth. Carbonate rocks mainly consist of highly soluble components, such as limestone, dolomite and gypsum. These components are easily dissolved by rainwater, which leads to the formation of crevices within the bedrock surface (Fig. 1). The number of crevices is proportional to the bedrock solubility[18], and crevices will create preferential flow paths that may contribute to enhanced regolith water loss by leakage, limits the retention of water in regolith, therefore, lead to high regolith water loss rate (RWLR)[18]. Moreover, since carbonate rocks have low contents of acid-insoluble components, only a small quantity of residue is left after dissolution[12], further limiting the retention of water.

The variation in the RWLR across the lithosequence will most likely result in remarkable differences in vegetation productivity. Areas dominated by soluble rock substrates are often characterized by large temporal variations in plant water availability that are induced by rainfall intermittency and droughts during periods without precipitation[19]. These droughts will have an important influence on the vegetation growth potential[20]. In contrast, for bedrock that highly resistant to solution, the accumulation of water improves because these bedrock types create an impenetrable barrier of residuals and in turn constrain water loss. In this type of environment, reduced water stress for plants over extended periods of drought may lead to increased overall productivity[21].

Here, we explore the factors controlling the vegetation productivity on carbonate rock through a case study in China to unravel the role of bedrock geochemistry, which remained unclear until present. A novel and key aspect of our approach is the consideration of the RWLR in order to evaluate the impact of the temporal variability in regolith moisture levels during dry spell events at the regional scale. This estimation allowed us to compare the relative strength of bedrock geochemistry versus other competing models (Table 1), and examine the bedrock geochemistry-RWLR-vegetation productivity relationships.

## Results

**Regional variations of predictor variables.** We selected 23 critical zone units (CZUs) are located along large climatic gradients (Fig. 2). From the south to the north of our study area, the mean annual temperature (MAT) decreases from 19.9 to 11.2 °C, whereas the annual precipitation (AP) increases from 739 to 2300 mm. CZUs also exhibited high variation in bedrock geocheimsty, with bedrock Si oxide concentrations ($BR_{Si}$) of 2.4% −49.3%, and bedrock Ca oxide concentrations ($BR_{Ca}$) of 4.8% −74.2%.

In this study, the RWLR was evaluated by variations in regolith moisture during dry spells, while the regolith moisture was represented by the temperature vegetation drought index (TVDI, see Estimation of RWLR in Methods for more details). During 2001–2010, 428 dry spell events were identified, with an average of 19 dry spell events in each CZU. Sixty-five percent of these dry spell events were <10 days in duration (Supplementary Fig. 1a) and occurred mainly during fall and winter (i.e.,

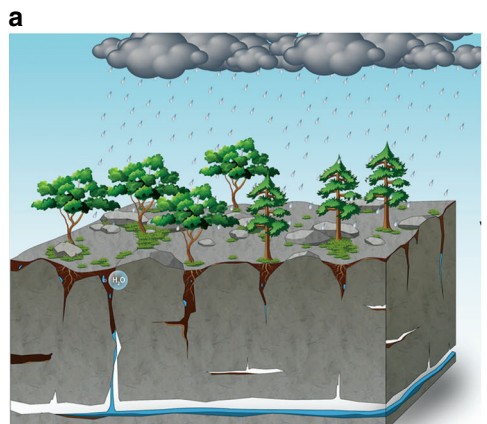
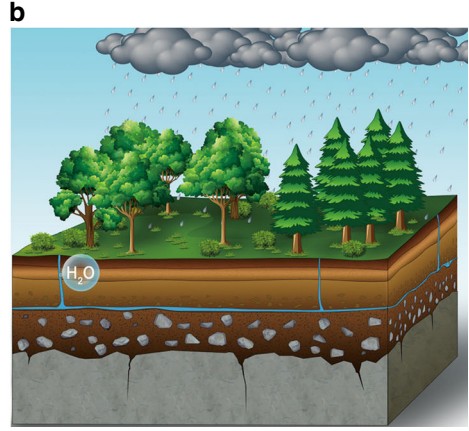

**Fig. 1 Illustration of our hypothesis. a** Karst critical zone structure; **b** non-karst critical zone structure; the bubble in the regolith represents the regolith water-holding capacity. We hypothesize that in the karst zone, the bedrock geochemistry can influence vegetation productivity through controlling the regolith water hold capacity: increased Ca concentrations correspond to increased limestone, which is highly soluble, so the bedrock develops an amount of crevices. These properties limit the retention of water and hence affect the vegetation productivity. We expect that in karst regions, by regulating the regolith water-holding capacity, bedrock geochemistry plays an important role in the spatial variability in vegetation productivity.

September–December) (Supplementary Fig. 1b). The values of the RWLR index and the bedrock Si and Ca concentrations ($BR_{Ca}$ and $BR_{Si}$) showed significant differences between carbonate and non-carbonate areas, while the other selected climatic parameters and contents of chemical elements did not exhibit significant differences between the carbonate and non-carbonate rock types (Supplementary Fig. 2).

**Relationships between NPP and predictor variables.** Among the selected soil and climate variables that could affect the net primary productivity (NPP) (Table 2), only the RWLR index and MAT showed significant relationships with the NPP. The RWLR index exhibited a negative relationship with NPP, whereas MAT was positively correlated with NPP. In particular, the RWLR have greater explanatory power as a predictor of the variation in NPP than other selected climate and soil variables (Fig. 3 and Table 2).

$BR_{Si}$ and $BR_{Ca}$ show significant correlations with the RWLR (Fig. 3 and Table 2). $BR_{Si}$ was negatively correlated with the RWLR, whereas $BR_{Ca}$ showed a positive relationship with the RWLR. Moreover, the information theoretic approach (ITA, see the Methods for more detail) confirmed that $BR_{Si}$ emerged as the best predictor of the RWLR.

**Influence of geochemistry on NPP through affecting RWLR.** The structural equation modeling (SEM) was used to quantify the indirect influence of geochemistry on NPP through affecting RWLR. To simplify the SEM, we did not include the independent variables of AP, Solar duration (SD), the Palmer Drought Severity

---

**Table 1 Set of hypotheses and related predictor variables used for analyzing the spatial variation in vegetation productivity, the selection of competing hypotheses account to previous studies.**

| Abbreviation | Description | Unit |
|---|---|---|
| Lithological control hypothesis (Fig. 1) | | |
| RWLR | Regolith water loss rate: variation in soil moisture during a dry spell | |
| Energy hypothesis: vegetation productivity determined by energy availability | | |
| MAT | Mean annual temperature[35] | °C |
| SD | Solar duration[35] | hour |
| Drought hypothesis: lack of precipitation or high evaporation result in drought stress, which limits vegetation growth. | | |
| AP | Annual precipitation[36] | mm |
| PDSI | Palmer Drought Severity Index[36] | |
| Soil fertility hypothesis: plant growth greatly affected by soil nutrients | | |
| Soil N | Soil nitrogen content[37] | % |

---

**Table 2 Relative variable importance (RVI) of each candidate variable when predicting net primary productivity (NPP) and the regolith water loss rate (RWLR).**

| Response variables | Candidate variables | AIC | RVI |
|---|---|---|---|
| NPP | RWLR | 270.11*** | 1.00 |
| | Soil.N | 289.03 | 0.41 |
| | SD | 289.32 | 0.24 |
| | MAP | 290.32 | 0.21 |
| | MAT | 283.05* | 0.18 |
| | PDSI | 290.2 | 0.18 |
| RWLR | $BR_{Si}$ | 48.43*** | 0.96 |
| | $BR_{Ca}$ | 48.43*** | (0.97) |
| | $BR_{Mg}$ | 58.73 | 0.44 (0.22) |
| | $BR_{Fe}$ | 59.52 | 0.22 |
| | $BR_{Al}$ | 55.11 | (0.20) |

Since $BR_{Si}$ and $BR_{Ca}$ are strongly correlated (Supplementary Table 1), we separated models into subset for RWLR in order to mitigate the collinearity in explanatory variables, with one subset included $BR_{Si}$ and the other included $BR_{Ca}$ are shown outside and inside the brackets, respectively. The Akaike information criterion (AIC) was obtained from single variable linear models (LMs), while the RVI was obtained from the information theoretic approach. For more detail, see Supplementary Tables 1 and 2.
Levels of significance are shown as: *$p < 0.05$, **$p < 0.01$, ***$p < 0.001$.

---

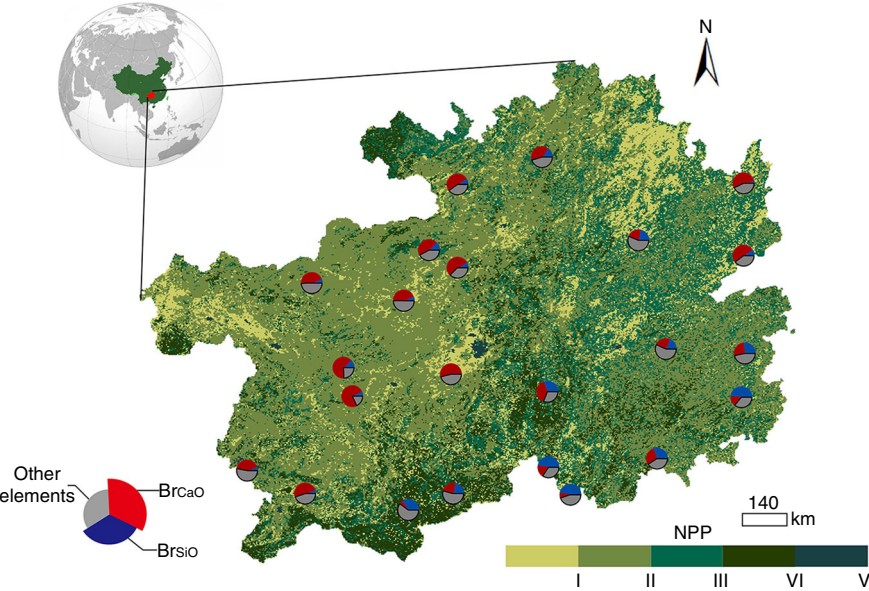

**Fig. 2 Spatial distribution of critical zone units in Guizhou.** The pie chart represents the location of the critical zone units considered in this study. The different colors within the pie charts represent the bedrock concentrations of the elements. Each critical zone unit has a meteorological observation station at the center with a radius of 20 km. the net primary productivity (NPP, g Cm$^{-2}$ yr$^{-1}$) is divided into five levels, i.e., I: 0.00–319.83, II: 319.83–513.40, III: 513.40–740.65, VI: 740.65–1060.48, V: > 1060.48.

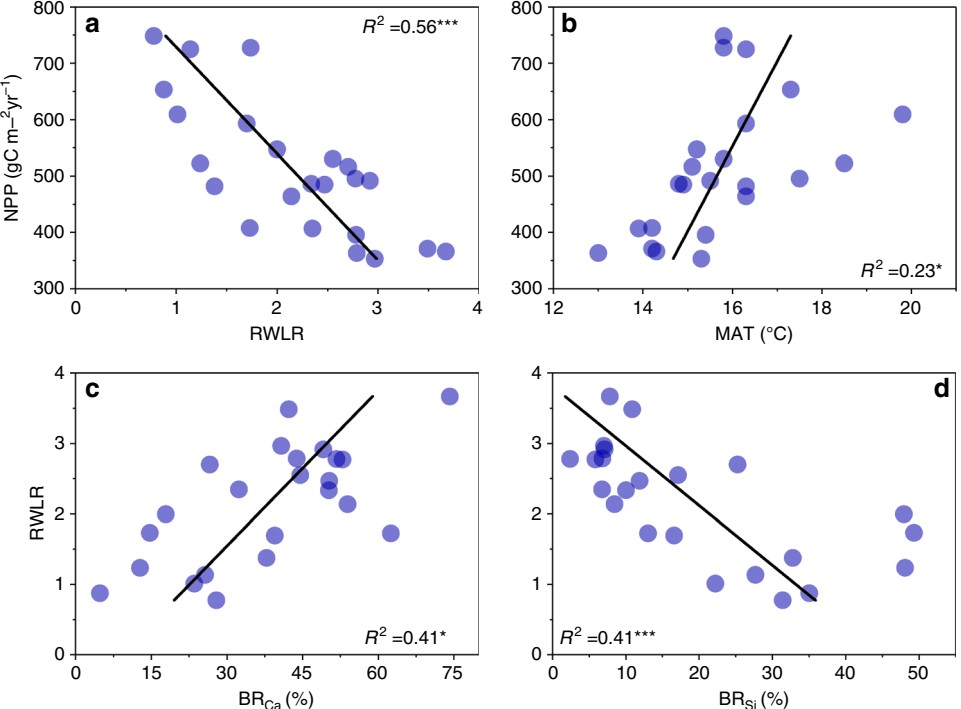

**Fig. 3 Linkages between NPP and predictor variables.** Relationships between NPP and **a** regolith water loss rate (RWLR), **b** mean annual temperature (MAT); the relationships between RWLR and: **c** bedrock calcium oxide content (BR$_{Ca}$), and **d** bedrock silicon oxide content BR$_{Si}$. Adjusted $R$-squared ($R^2$) and levels of significance are shown (*$p < 0.05$, **$p < 0.01$, ***$p < 0.001$).

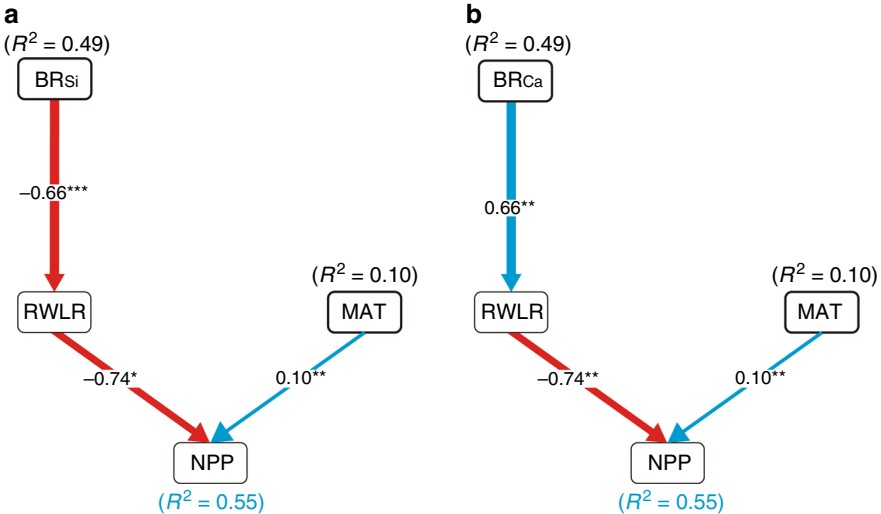

**Fig. 4 Structural equation models of vegetation productivity.** Two separated SEMs for the indirect effects of bedrock chemical elements (left: BR$_{Si}$, right: BR$_{Ca}$) on NPP. The selection of explanatory variable terms in the results of single LMs and previous knowledge (Table 1). The blue arrows indicate positive effects, and the red arrows show negative effects. Standardized regression coefficients are given, with the thickness of the arrows expressing the size of the standardized regression coefficients. The $R^2$ values of each model (black) and the total effect of response variables (blue) are given between the brackets. Adjusted $R$-squared ($R^2$) and levels of significance are shown (*$p < 0.05$, **$p < 0.01$, ***$p < 0.001$).

Index (PDSI) or soil variables in our model, given that these variables did not show significant relationships with the NPP (Table 2). Each variable included in the SEM exhibited either a direct or indirect effect on the NPP and contributed to the simplification of the model.

Given BR$_{Si}$ and BR$_{Ca}$ showed strongly negative correlation (Supplementary Table 1), to mitigate the problematic effects of collinearity, they were separated into two SEMs (Fig. 4). The separated SEMs performed well, with both explaining 55% of the variations in NPP. The results of the SEMs were consistent with the output from the ITA analysis, thereby showing that bedrock element contents have greater explanatory power than the MAT as a predictor of NPP. BR$_{Si}$ and BR$_{Ca}$ explained 49% of the variations in NPP, respectively, while MAT explained only 10%. The indirect effect of bedrock geochemistry on the NPP through influences on the RWLR was identified by the SEMs. BR$_{Si}$ and BR$_{Ca}$ were highly correlated with the RWLR, which exhibited significantly negative and positive relationships, respectively. Both SEMs showed that the RWLR was negatively correlated with NPP.

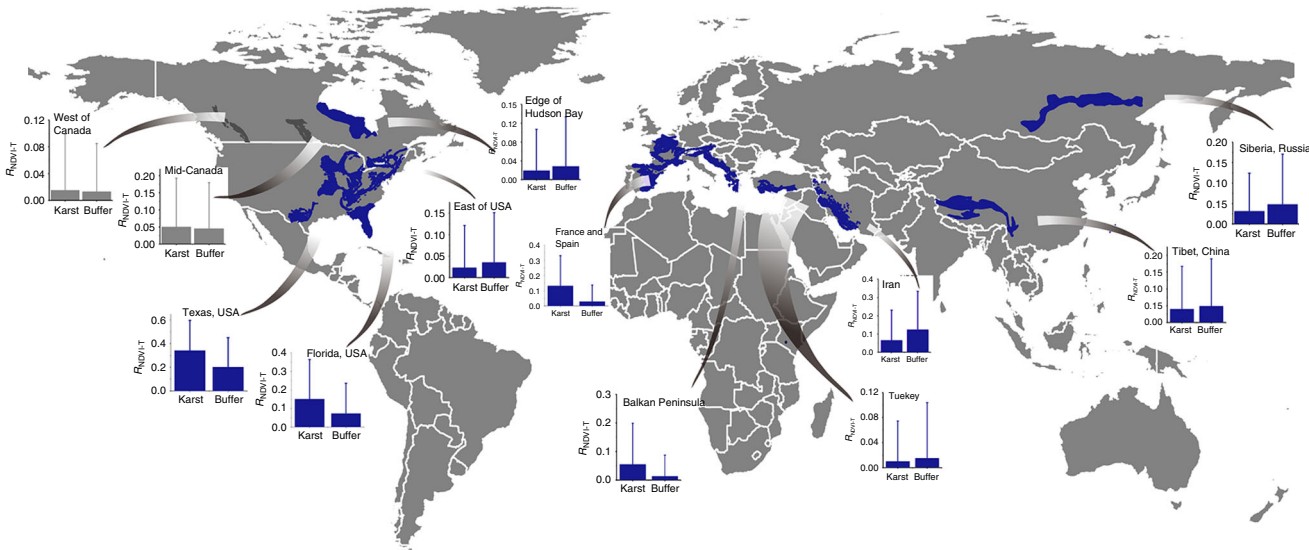

**Fig. 5 Locations of the selected 12 largest karst regions.** The selected karst regions are located at mid-latitudes (i.e., 30° to 60°N) in the Northern Hemisphere. The dark gray, and blue parts are the carbonate rock areas. The relationship between the NDVI and the growth season (April–September) mean temperature ($R_{NDVI-T}$) was calculated at the pixel level. Subsequently, the difference in $R_{NDVI-T}$ between the inner and outer areas of carbonate rock buffer zones (with a width of 50 km) using a Mann–Whitney Wilcoxon test. Bars represent means for each treatment and error bars are 95% confidence intervals of the mean. Blue graphs indicate that the $R_{NDVI-T}$ value in the inner carbonate rock area is significant different with the outer area, dark gray graphs indicate that there are non-significant difference in $R_{NDVI-T}$ between both areas.

**Vegetation activity–temperature relationships in global karst**. The mean normalized difference vegetation index (NDVI) from April to September was used as a measure to assess vegetation activity across the 12 largest karst regions located at mid-latitudes of the Northern Hemisphere. The results showed that vegetation activity exhibited different sensitivity to interannual temperature variations in these karst regions compared to the surrounding areas, with the exception of the karst zone located in the middle and western parts of Canada (Fig. 5). Out of the 12 karst regions, 6 showed significantly low correlations between the NDVI and mean temperature for the growing season ($R_{NDVI-T}$), while 4 of them showed high correlations compared to those surrounding areas, suggesting that carbonate could alter the effect of temperature on vegetation growth.

## Discussion

With a comprehensive, broad-scale assessment of the influence of bedrock properties on the regional variations in vegetation productivity, we determined that bedrock geochemistry can strongly regulate vegetation productivity at the regional scale by influencing the regolith water storage capacity. More importantly, even with the marked variations in the climatic and soil conditions across the study area, the bedrock geochemistry exhibits a major explanatory role for the variations in vegetation productivity, which highlights that the role of bedrock geochemistry is more important than so far assumed.

The strong bedrock geochemistry-RWLR-vegetation productivity associations could be elucidated from several possible mechanisms. The bedrock geochemistry might be linked to its mineral composition in a way that responsible for solution process, which thus influence the regolith water-holding capacity[16,22]. Solution-resistant minerals are quite different in the Ca-rich and Si-rich bedrock, resulting in differentiated water-holding capacity substrate and consequently differentiated vegetation growth potential.

Other possible lithological controls on the regolith water-holding capacity might act by influencing the regolith thickness[23–25]. In the karst regions, high Ca concentrations in bedrock might result in thin regolith, which on its turn might correspond to increases in calcium carbonate concentrations, resulting in low-regolith formation rates for bedrocks with limited acid-insoluble components, as only a small amount of insoluble residues are derived from dissolution[26]. Moreover, regolith loss because of underground leakage will further limited the remaining amount of the regolith in these environments[27]. Thus, the positive correlation between RWLR and Ca concentrations may reflect the differences in regolith thickness.

In addition, the positive correlation between the regolith water-holding capacity and bedrock Si concentration could be explained by the clay content. Si-rich bedrock derived regolith is typically characterized by high-clay contents. These high-clay contents improve the regolith water-holding capacity, as they create a regolith structure with smaller pores, and therefore, water can be held at higher suction pressures[23]; or because clays create a less penetrable layer that restricts the rate of water infiltration[24]. The models, including BR$_{Si}$ and BR$_{Al}$, show the third best explanatory power for the RWLR (Supplementary Table 2b), which may be evidence of such inference.

The bedrock-vegetation connection could also be elucidated from nutrients. Many mineral nutrients that are extremely important for plant growth (e.g., P and Fe) are derived from bedrock, and their availability may be controlled by the chemical and physical processes at the soil-bedrock interface[25,28]. This phenomenon is particularly important in karst regions, where, for example, the P availability is strongly controlled by the CaCO$_3$ content in regolith[29,30], as CaCO$_3$ can reduce the P solubility by producing Ca phosphates[31]. Hence, as the P limitations are widespread in areas characterized by CaCO$_3$-dominated sedimentary rocks, this is a key factor limiting vegetation development. However, a lack of P availability data along the lithosequence restricts the current level of knowledge, and therefore future experimental research on designing fertilization experiments is needed to investigate the functional relationship between P availability and bedrock compositions.

By regulating the hydrologic properties of regolith, bedrock composition can play a fundamental role in vegetation growth. It is reasonable to believe that such lithological regulation is strong enough to change the response of vegetation to climatic factors. More importantly, the different temperature sensitivities of karst vegetation show that the bottom-up effect of bedrock could presumably have widespread significance in karst regions worldwide.

The different temperature sensitivity of karst vegetation might link to multiple mechanisms. We suspect that the typically low-regolith water-holding capacity of these systems significantly enlarges the impacts of drought events on the vegetation development, and therefore change association of vegetation productivity to temperature. Moreover, it is noteworthy that the influence of land use, which can interact with role of bedrock, strongly change the response of vegetation to climate change. While karst is usually characterized by thin soil layer and a number of sinkholes underground[32], irrational land use could cause serious soil erosion, exposing the bedrock to the surface. The resulting rocky landscape could have a different association of vegetation productivity to temperature. In addition, the bedrock could be involved in amount of ecological processes and its influence is likely common worldwide, about which our knowledge is still very limited.

Since bedrock have the potential to obscure or distort the influence of climate, our ability for predicting vegetation activity under warming climate will benefit from empirical research of lithological control. For example, in Guizhou Province, it is reasonable to expect that increasing temperature would have stronger influence on karst vegetation, since they suffer more drought stress than non-karst regions. However, the influence of bedrock has not been widely addressed, therefore, we hope the evidence presented here will provide motivation for other ecologists to explore the bedrock-vegetation link in other karst systems.

Although our study highlights the impact of bedrock geochemistry on vegetation productivity, we do not intend to deny the importance of climate. The dissolution and weathering of bedrock is not only determined by its mineral compositions, but also depend on climate. We have to admit that, due to the limitation of direct measurement of regolith thickness and water-holding capacity as well as observation of ecological processes at regional scale, the bedrock geochemistry-regolith water-holding capacity-vegetation productivity connections can only be elucidated from statistical analysis rather than direct observation in this study. Moreover, as we focus on a regional level impact of bedrock geochemistry, we have not discussed the great heterogeneities in belowground future, climate feature, and vegetation composition within the karst critical zone in our study region. More explicitly assess the role of bedrock will benefit from long term critical zone monitoring and experimental research worldwide.

In conclusion, our study emphasizes that bedrock geochemistry has a great potential to influence plant growth in kart areas through controlling the regolith water-holding capacity. Hence, these results indicate that the role of bedrock geochemistry in vegetation productivity could have been underestimated. Despite the uncertainties, the critical zone approach employed in this study provide evidence about importance of bedrock on vegetation, as the functioning of terrestrial ecosystems is determined by a wide range of processes with complex interactions across the atmosphere, biosphere, pedosphere and lithosphere, all these components need to be considered to obtain a full and detailed picture of any studied terrestrial ecosystem. Therefore, an understanding of vegetation productivity and associated drivers can be obtained from coordinated multidisciplinary scientific research approaches, which are based on the critical zone framework as presented in this paper.

## Methods

**Regional settings**. The study area covers the entire Guizhou Province (N24° 30′–29°13′, E103°1′–109°30′; Fig. 2). This area is 128,480 km² and is characterized by numerous karst geological formations, accounting for 64% of Guizhou Province's lithology, which contains a variety of different carbonated rock types. The remaining 36% of the lithology (mainly present in the southern, northern and southeastern parts of Guizhou) consists of igneous and metamorphic rocks, which are typically rich in silicon (Si) and have low calcium carbonate contents. The study region is characterized by a subtropical humid monsoonal climate with a MAT of ca. $11-20\,^{\circ}C$ and mean AP of ca. $730-2300$ mm. The precipitation is seasonally variable, with ~75% of the AP occurring during the summer and autumn (June–November).

**Data compilation**. To study the vegetation-regolith-bedrock links, 23 CZUs were established (Fig. 2 and Supplementary Fig. 3) with a radius of 20 km and a meteorological observation station at the center of each zone. The selection of sites was based on the quality of precipitation data. To estimate the regolith water-holding capacity of each site, it was necessary to investigate the precipitation patterns and estimate the variation in soil moisture during dry spells, so each CZU should have a meteorological observation station at the center. There are 71 meteorological observation stations in Guizhou Province, but only 30 of them have precipitation data covering a time span of 10 years (i.e., 2001–2010). We also excluded the sites on which the associated meteorological observation was relocated during 2001–2010, as well as sites characterized by <10 identifiable dry spells, resulting in a final selection of 23 sites in total, which satisfy our study design: the selected sites were distributed evenly throughout Guizhou Province, including different bedrock types and along a wide climate range; the sample size was appropriate in order to carry out all the presented statistical analyses because we treated each site as a sample, which makes our statistical approaches robust, and hence, our results are reliable. This particular CZU area extent was chosen based on the recommendation of Liu et al.[33], as the meteorological station data are representative of the entire zone, because homogeneous climatic conditions within each CZU can be assumed.

The 23 CZUs are well distributed across the entire study area (i.e., Guizhou Province, Fig. 2). Fourteen of these CZUs are located on carbonate rock, whereas nine are on clastic rock. The remote sensing retrieved terrestrial annual NPP values were obtained from Resource and Environment Data Cloud Platform with a spatial resolution of $1 \times 1$ km (http://www.resdc.cn/Default.aspx), which was estimated from the light energy utilization model (GLM_PEM)[34]. This dataset was used to obtain an approximate estimate of the regional vegetation growth for each CZU. The time span for both the climatic and NPP data was from 2001 to 2010.

**Predictor variables for NPP**. We used a total of seven predictor variables (Table 1) in order to explain the vegetation productivity. These variables were selected since they are related to the hypotheses of this study and widely considered potential drivers of vegetation productivity in Guizhou[35–37]. More details about these variables were provided below.

We used meteorological station data (available at http://data.cma.cn/), which included daily temperature, precipitation and solar radiation covering the period 2001–2010. The associated climate variables (MAT, SD, AP, PDSI) were calculated for each CZU.

In each CZU, three sites were selected for bedrock and soil sampling within each lithologic unit (Supplementary Fig. 3), and the delineation of the geological system was based on the 1:50,000 scale geologic map from the National Geological Archives of China, http://www.ngac.org.cn). In total, 204 sampling sites were established. To avoid artificial disturbance effects, the selected sites were kept as far away from farms, towns and cities as possible. All soil and bedrock samples were taken from fresh road cuts. To reduce the influence of fire events and grazing activity during evaluation of importance of bedrock geochemistry to vegetation productivity, the selected road cuts were covered by intact and undamaged vegetation. At each site, three profiles were sampled, each of which were separated by $5-50$ m, depending on the length of the road cut. A ring-knife was used to collect soils from $0-30$ cm depth and fresh bedrock samples were collected using a geological hammer.

In total, 612 bedrock and 607 soil samples were collected (five soil samples were not obtained due to a lack of significant soil coverage). In the laboratory, composite samples were made by thoroughly mixing the replicate soil samples (from the same stratigraphic unit and CZU) before being oven dried (40 °C, 72 h) and sieved (2 mm mesh size to remove plant material and stones). Subsequently, each sample was crushed and milled to 100 μm to allow for soil chemical measurements. The total soil nitrogen (soil N) was measured by a standard Elemental Analyzer (Vario EL, Germany). All bedrock samples were crushed and powdered to 50 μm using a three head grinding machine (XPM-φ120 × 3, China). Replicate bedrock samples (same stratigraphic unit and CZU) were mixed and then fired (550 °C, 12 h). Concentrations of major bedrock elements were measured by X-ray fluorescence spectrometry (M4 TORNADO, Germany). Soil N contents and the major bedrock

elemental concentrations were weighted by area fractions of the different stratigraphic units present within each CZU.

**Estimation of RWLR.** In the karst regions, the hydrological function of substrate is characterized by high-spatial heterogeneity[38]. More precisely the regolith water-holding capacity is not only influenced by regolith thickness and topography, but also depends on the network of fractures and fissures (Fig. 1). Thus, even in the CZU dominated by geochemically homogeneous bedrock, the variations in the RWLR can be great within a CZU. Hence, the relationship between bedrock geochemistry and RWLR is difficult to detect through direct measurements over a small spatial scale (e.g., considering detailed on-site monitoring).

Nevertheless, large scale surveys of regolith water dynamics can help to determine whether bedrock geochemistry is an important regulator of the RWLR. In this study, we assessed the RWLR across the different CZUs using a satellite-based index. We quantified the variations in soil moisture during dry spells, assumed that a high RWLR corresponded to high variations in soil moisture. The TVDI that has been widely used for soil moisture monitoring is used as a surrogate variable of soil moisture[39,40]. More importantly, previous studies showed that TVDI was highly correlated with soil moisture in karst regions located in southeast China[41,42]. In this study, variations in the TVDI during dry spells were used to represent the RWLR (Supplementary Fig. 4).

TVDI is based on an empirical parameterization of the relationship between land surface temperature and vegetation indices[43]. As TVDI is relatively insensitive to precipitation, it is appropriate to adopt this index to estimate the soil moisture variations related to the RWLR. More precisely, TVDI was calculated as:

$$\text{TVDI} = \frac{T_{\text{obs}} - T_{\text{w}}}{T_{\text{d}} - T_{\text{w}}}$$

$$T_{\text{d}} = a + b * \text{NDVI}$$

$$T_{\text{w}} = c + d * \text{NDVI}$$

where $T_{\text{obs}}$ is the surface temperature at each pixel; $T_{\text{w}}$ represents the minimum surface temperature under the given vegetation conditions, while $T_{\text{d}}$ represents the maximum surface temperature, which is estimated from the edge of the $\text{NDVI}/T_{\text{obs}}$ space (3). Therefore, $T_{\text{w}}$ and $T_{\text{d}}$ are linear functions of NDVI, the $a$, $b$, $c$, and $d$ associated parameters were estimated at the pixel level, and hence, high TVDI values indicate high-surface water deficits.

Furthermore, as the monitoring of soil moisture changes during dry spells requires fine temporal resolution data, the satellite images used to calculate the TVDI were obtained from the MODIS–Terra sensor (i.e., Ts: MODLT1F, 5-day scaled at 1 km resolution; NDVI: MODND1F, with 5-day interval at 500 m resolution. http://www.gscloud.cn/), covering a time period from 2001 to 2010. The Ts and NDVI products includes atmospheric corrections to eliminate the influence of background noise. To match the pixel size of the NDVI map with the Ts map, the 500-m resolution of NDVI images were resampled to 1 km, using the nearest-neighbor-interpolation technique. The TVDI images of Guizhou Province of China were analyzed using ENVI 5.1 software, and then each CZU was extracted using ArcGIS. If the qualities of the NDVI or Ts images were unacceptable for the TVDI estimation, they were excluded.

For each CZU, the RWLR was calculated as the average variation in TDVI during dry spells (Supplementary Fig. 4). In this study, a dry spell was defined as the period of 5 or more consecutive days with no precipitation:

$$\text{RWLR} = \sum_{i=n} \frac{\overline{\text{TVDI}_{b,i} - \text{TVDI}_{a,i}}}{\Delta T_{ab,i}} \times 100$$

$n$ represents the number of observable dry spells from 2001 to 2010; $i$ represents the $i^{\text{th}}$ dry spell, $a$ and $b$ represent the $a^{\text{th}}$ and $b^{\text{th}}$ TVDI ($b > a$), $\Delta T_{ab}$ represents the time interval between $a$ and $b$.

**Statistical analysis.** To compare the importance of the RWLR in predicting the NPP with other variables, and to evaluate the predictive power of the bedrock concentrations of the major elements for the RWLR, the ITA[44] was performed. This approach allowed us to determine the relative variable importance (RVI) of each explanatory variable. All possible combinations of explanatory variables were tested. The best model subset for NPP and RWLR was identified by summing the Akaike weights of the highest ranked models until the value exceeded 0.95. The RVI for each of the candidate variables was calculated by summing the Akaike weights for all models in which the variable occurred in the best model subset.

To avoid collinearity among variables, we pre-selected the candidate variable based on our hypothesis and the performance of the prediction, which was estimated using a linear model (LM). Wherever two variables exhibited strongly collinearity (Pearson's $|r| > 0.70$)[45], we excluded the variable showing weaker (greater Akaike information criterion (AIC)) with the response variables. One exception was $\text{BR}_{\text{Si}}$ and $\text{BR}_{\text{Ca}}$, which were strongly correlated (Supplementary Table 1) but we did not exclude either because both showed a significant relationship with RWLR and the AIC values were very close; and they even showed a high collinearity, but they might influence the RWLR via different mechanisms.

Therefore, we separated the RWLR models into two subsets, with $\text{BR}_{\text{Si}}$ excluded in one model subset, and $\text{BR}_{\text{Ca}}$ excluded in the other subset.

The LMs were conducted using $\text{R}_{3.1.0}$ software[46], and the model selection was performed using the "dredge" function in R package $\text{Mu}_{\text{MIN}}$[47].

**Structural equation model of vegetation productivity.** To explore the interactions between bedrock geochemistry, RWLR, and NPP, and assess the indirect influence of bedrock geochemistry on NPP, a SEM[48] was developed to understand the relationships among bedrock geochemistry, the RWLR, climatic factors, soil nutrients and NPP. The use of a SEM allows for the direct and indirect effects of candidate variables to be considered. Furthermore, SEM can be used to test whether the integral model is statistically acceptable. In our SEM, the RWLR and vegetation productivity were treated as response variables, whereas the bedrock geochemistry, and climatic and soil variables were treated as explanatory variables. First, we designed an SEM to investigate the relative importance of bedrock geochemistry on vegetation productivity through regulation of the RWLR. Additionally, we assessed the influences of climate and soil variables on the NPP, as well as the relationships with the RWLR.

Since this study is characterized by a rather small sample size (23 CZUs), a model simplification is required. We considered only those explanatory variables showing significant relationships with the response variables in our SEM analysis. An additional component for the SEM was developed to evaluate the residual correlations and modification indices after model's execution. The root mean square error of approximation (RMSEA) and the comparative fit index (CFI) were used to evaluate the goodness of fit. The final SEM was chosen when the following criteria were met: $p$-values of $\chi^2$ and the goodness of fit test $p > 0.05$; CFI > 0.9, and lower 90% confidence intervals (CIs) for RMSEA < 0.05 (ref. [18]). The SEMs were calculated using the R packages "sem"[49] and "lavaan"[50].

**Calculation of the NDVI-temperature relationship.** To estimate the influence of bedrock on changes to the response of vegetation activity to temperature variability, 12 largest karst regionss from mid-latitude bands (30°–60°) of the Northern Hemisphere were selected to detect their temperature variation signals of vegetation productivity. We chose this area because: first, the mid-latitudes in the Northern Hemisphere exhibits obvious climate seasonal variation, and climate change will most probably have a significant influence on the vegetation productivity[51,52]. Second, A total of 61% of the world's karst regionss are distributed in this zone[53]. Moreover, the carbonate rock areas located in deserts were discarded from further analysis, as the NDVI-values highlighted the lack of any significant vegetation development in these areas.

The temperature variation signal of vegetation productivity is calculated as the correlation coefficient between NDVI (obtained from Global Inventory Modeling and Mapping Studies, spatial resolution: 0.5°; biweekly data from 1982 to 2011)[54] and growing season mean temperature (April–September, obtained from Climate Research Unit[55], the spatial resolution and temporal range are consistent with NDVI) ($R_{\text{NDVI-GT}}$) at the pixel level. To assess the influence of bedrock on the relationship between vegetation activity and temperature, we employed a buffer zone approach. We compared the $R_{\text{NDVI-GT}}$ within carbonate rock areas and buffer zones outside the carbonate rock areas boundaries by Mann–Whitney Wilcoxon test. The width of these buffer zones is 50 km (1 pixel), because this width minimizes the influence of climate and results in similar numbers of pixels in the inner and outer areas.

## Data availability
The authors declare that the source data supporting the findings of this study are provided within the paper.

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

## Acknowledgements
We thanks Naiqing Fan, Yaomin Qin, Xu Liu, and Xiguang Yang for their help in remote sensing data treatments, and Jie Liu, Canfei Hu and Ridi Li for their support in collecting bedrock geochemistry data. This study was jointly financed by the Sino-UK Critical Zone Programs of National Natural Science Foundation of China (grant no. 41571130044), and the National Environmental Research Council of the UK and the Newton Foundation (grant no. NE/N007603/1 and NE/N007530/1).

## Author contributions
Z.J., J.M., and S.G collected and analyzed the data. H.L. designed the project and this study; Z.J. wrote the manuscript; H.L., H.W., J.P., J.M., S.G., T.Q., X.W., Z.S. revised the manuscript. All the authors discussed the results and commented on the manuscript.

## Competing interests
The authors declare no competing interests.
