## [Peer Review File · Nature Communications]

Reviewers' comments:

Reviewer #1 (Remarks to the Author):

Dear authors and Editor,

I have now read the manuscript entitled "Bedrock geochemistry impact exceeds climate effects on vegetation growth in karst regions by Jiang and co-authors. The manuscript is very well written, and quite interesting. I have some minor comments and questions, but in general question the extrapolation of results to larger scales and thus the very strong overall conclusions (as reflected in the title). Below, I list some comments/questions in the order in which they appear throughout the manuscript.

Title: the statement that geochemistry impacts exceed climate effects is quite strong and, in my opinion, not fully supported by the results.

Line 23: Earth with capital letter.

Line 29: Should "Karst" be capitalized?

Line 34: Write "explained".

Line 38: Sounds better to write "...; for 16 of these regions, temperature showed...".

Line 122-124. Although the RVI for RWLR is indeed higher than for MAT, in explaining variation in NPP, it is only very slightly so (and not quantitatively assessed), why the statement that RWLR is more important than MAT is not fully supported (in my opinion).

Line 148: But in Figure 4, the correlations between RWLR and NPP are depicted as positive (i.e. no negative sign for correlation coefficients and blue-coloured arrows). What is correct?

Line 157-159: Better to write "Out of the 17 karst zones, 10 showed a significant correlation...", but also give the direction of the correlations. Further, looking at Figure 5, I am not entirely convinced that this is a good argument for the overall conclusion. There will always be variation around a mean, and a few of the 17 sites are quite close to (and below) the global average and 1 site is above. Hence, without any further quantification of the strength of these results (I mean, it is just visually assessed), I question their support for the strong conclusions taken by the authors (e.g. Line 214-224).

Figure 4. Maybe a lighter blue would be better, as the current colour looks close to black. See also comment above about directions of relationships (Line 148).

Reviewer #3 (Remarks to the Author):

This is a review of NCOMMS-19-13879: "Bedrock geochemistry impact exceeds climate effects on vegetation growth in karst region" for Nature Communications.

When composing your report, the following questions might assist you in writing an incisive, well-justified review.

****What are the major claims of the paper?**

This paper reports results of an analysis of the effects of lithology on ecosystem productivity in karst terrain. It focuses on sites in Guizhou province in China, but later extends the analysis to karst terrain

around the world.

The main claims are: bedrock concentrations of Si and Ca explain variance in "regolith water loss rate", which in turn explains variance in net primary productivity. Hence, Si and Ca concentrations explain variance in NPP. Finally, the relative lack of correlation between vegetation and temperature in karst terrain around the world (relative to everywhere else) is suggested to imply that lithological control on vegetation in karst terrain is widespread.

**Are they novel and will they be of interest to others in the community and the wider field?

To my knowledge, the claims are novel and would be of interest to a broad swath of biogeographers, critical zone scientists, and karst region hydrologists.

***If the conclusions are not original, it would be helpful if you could provide relevant references.

To my knowledge, the claims are original.

***Is the work convincing, and if not, what further evidence would be required to strengthen the conclusions?

The work is not convincing.

Overall, I think the authors did a good job of setting up a problem worth solving, and importantly, they outlined a reasonable, testable hypothesis about the different drivers of vegetation productivity in different lithologies — specifically, karst versus silicate bedrock.

However, it's impossible to determine whether or not they have gathered the data and conducted the analyses that are needed to test the hypothesis.

First, there is no rationale given for the sites they chose. Why those sites? Is there a study design that these sites satisfy?

Bigger problem: After reading the manuscript, the methods, and the supplemental file and then studying all of the figures and tables as carefully as possible, I still do not know how the authors measured any of their key variables. NPP? RWLR? BrSi? BrCa?

There is a sense that NPP was maybe measured using remotely sensed NDVI. But how was that done? And there is a sense that RWLR is measured using remotely sensed TDVI. But how exactly? And since these remotely sensed indices are themselves calculated from different combinations of Landsat bands, doesn't that make it inevitable that they will be strongly correlated (because the same variables are used to calculate both the response and driving factors in the relationship)? And if that's the case, since the NPP-RWLR connection is a linchpin of the paper, doesn't that pretty much call the entire analysis into question.

With the BrSi and BrCa, again, I am at a loss for how these were measured. There is difficult to understand conceptual diagram in the supplemental file (Figure S4) that seems like it may provide hints. But I just do not know. Were Si and Ca ratios actually measured somewhere?

Beyond the measurements themselves, which are not even reported anywhere in the paper or supplemental files that I had access to (making it impossible to reproduce the analysis), there is the data analysis, which is likewise difficult to follow. I go into some detail about that below, when I

address the appropriateness of the statistical measures. Bottom line is, there is no rationale provided for the choices the authors made about how they chose to look for dependencies and codependencies in the data. To be honest, my sense after after digesting the paper is that they indiscriminately applied a bunch of R modules that sounded to them like they might work and then just treated the individual analyses as black boxes, without really considering whether they were appropriate or not.

In addition to my concerns about the lack of a clear explanation of how things are measured, a lack of data reporting, and a lack of rationale for why they were analyzed in the way they were, I am surprised that the authors did not spend any time at all discussing what would seem to be an 800-lb gorilla in the room on this question: land use. It seems like it would be a major confounding factor in the analysis of vegetation patterns, especially in China, where deforestation and conversion of landscapes to agriculture have completely altered the ecosystem. This is also likely a confounding factor in the global analysis, but there are even bigger problems with that in my mind.

Regarding the global analysis, the chain of inference that the authors expect us to follow here is: 1) the correlation coefficients in 16 out of 17 karst terrains they analyzed are lower than the global average for correlation between NDVI and mean annual temperature, 2) this means that NPP is controlled by lithology in these terrains. It just does not hold up. One does not follow from the other. There is the statistical issue of how much lower do the coefficients need to be? They do not look very different to me (Figure 5). If they were reported with their uncertainties, is there overlap? We do not know because there is not a proper uncertainty analysis on these correlation coefficients.

In summary, from top to bottom — across all the analyses presented here I am unconvinced. The authors had a good idea, but I am not sure what they did to test it. If I follow them correctly, then much of what they did may be wrong, with artifactual correlations (NPP vs. RWLR) and potentially major confounding factors (land use) unaccounted for and not discussed at all in the text.

***On a more subjective note, do you feel that the paper will influence thinking in the field?

If the claims were sufficiently supported, I think the work would indeed influence thinking in the field.

***Please feel free to raise any further questions and concerns about the paper.

Overall, the writing in this manuscript is mostly OK, but needs a careful editorial pass to remove some problems like this one on line 230: "...Guizhou Province's lithology, which contains a variety of different ****carbonated**** rock types."

The methods section is grossly inadequate. There is virtually no explanation here of how ANY of the main variables were measured.

For example, the method for measuring NPP — one of the two key response variables in the study — is never explained for their study sites in China. For the global analysis, it looks like maybe NDVI is used as a proxy for vegetation. Here is the text on that: "The temperature variation signal of vegetation productivity is calculated as the correlation coefficient between NDVI (obtained from Global Inventory Modelling and Mapping Studies...)." Is this how NPP was estimated at the China sites? We do not know. We do not even see a data table containing the NPP values. (At bare minimum there needs to be a data table with all of the main variables used in the statistical analyses somewhere in the supplemental file if not in the main text.) Because they do not say and they do not report the basic data needed to understand the work. And if NPP is estimated from the global NDVI data, is that even appropriate? Is it at an appropriate scale?

An explanation of how they quantified the other key response variable is also not given. The authors

introduce this “regolith water loss rate” thing out of nowhere as the thing that is controlled by lithology and that in turn controls vegetation, but they never say how it is measured. Nor are values for RLWR reported anywhere. The only reference I can find to how it is measured is the quote: “ RLWR is estimated by evaluating the temporal variation in regolith moisture during dry spell events at the regional scale.” How is regolith moisture measured? From TDVI? But isn’t that just a combination of Landsat bands? And if that is the case, is it not inevitable that there will be a strong correlation between this so called RWLR and NPP, since one is measured from TDVI and the other is presumably measured from NDVI?

There is also no explanation of how they measured Si or Ca or Mg or Al or Fe concentrations. Are there actual measurements of bedrock geochemistry somewhere? If so why are they not reported? It says soil N is the average of “N-values from the three sampling points.” So the authors collected 3 soil samples from each CZU? Why is the sampling protocol not described somewhere? Maybe they also sampled bedrock? What about the N analysis procedure? How was this done? I am totally lost as to what the authors did.

The supporting information file is grossly inadequate in the following ways:

First, the lack of detail provided in the methods summary was not overcome here in the supplementary information. If the authors have shortchanged the methods section in the main text because of length constraints or some other reason, it should be corrected here. But there is no additional text. Just two new poorly explained tables and five new poorly explained figures. And worst of all, they raise more questions than they answer.

The figure captions are far too short, failing in all cases to adequately explain the contents of the figure. This is especially true for the concept figures S4 and S5. For example, in figure S4, no information is given about how the supposed lithologic boundaries in the CZUs were identified. Also, precisely what was sampled at the triangles? Bedrock? Soil? Remotely sensed NDVI?

The tables also have far too little explanation. For example, it is not clear from table S2 which model is being used. GLM? Mann Whitney Wilcoxon? SEM?

It seems that one solution would be to write a brief section about each figure and table. What do the results in each of these pieces show? What part of the paper do they support? Why did you use this test versus another? What are the pitfalls if any of these techniques?

***We would also be grateful if you could comment on the appropriateness and validity of any statistical analysis, as well the ability of a researcher to reproduce the work, given the level of detail provided.

Where to start? Perhaps with the summary: The statistical analyses were largely inappropriate and unreasoned. A snowstorm of techniques was used, but there is not a clear rationale given for any of them. Why is GLM more appropriate than a more straightforward multiple linear regression model? What can we learn from these models? Isn’t SEM used with social sciences datasets? Why is it useful here? What are the limitations of these approaches and why are they well suited to analyzing the data presented here (or rather not presented here, since there is no data table anywhere)? Is it appropriate to test every variable you have as a potential explanatory variable? What about the well know problem of multiple comparisons, where the overall false positive rate goes way up? By the way— because the data are not presented anywhere, it would be impossible for anyone to reproduce this work.

I am also concerned about the analysis of the global data base of karst terrain. It seems like it is reckless to espouse the chain of inference that the authors seem to want us to follow here. The lack of correlation between NDVI and T means lithology controls NPP in other landscapes? It’s not a chain of

inference I would be willing to put any stock in. Moreover, it is not clear that the correlation coefficients are even all that different from the "global average." An uncertainty analysis is warranted here.

Because of the lack of a clear rationale for each analysis technique, my overall impression was that the authors did not actually know what they were doing on the data analysis. Rather than follow a reasoned approach to sorting out the dependencies, including the collinearities in the explanatory variables, they threw every approach they could at the problem. It's almost as if they just decided to try everything they could find in R modules both arcane and more standard and then treated each module as essentially a black box.

Reviewer #4 (Remarks to the Author):

This is a very interesting and well-written paper that brings up new insights and challenges existing knowledge relating ecosystem productivity with abiotic factors that constraint and determine this function. This alone justifies its submission to a journal like Nature Communications. The authors propose very provoking and revolutionary hypothesis to explain and define ecosystem productivity using the critical zone concept and goes beyond current understanding of ecological processes. However, I find that to be able to solidly ground their proposed hypothesis, the authors need to be more thorough in the use of ecological theory, and provide a better theoretical explanation to their hypothesis. The authors jump very quickly to a general hypothesis from limited ecological data that relates to average climatic conditions and soil information. The authors extend their analysis to a global extrapolation, but the same problems remain in their global consideration. In synthesis, I find that the use of average conditions, as well as only a limited number of climatic and pedological descriptors precludes general descriptions of ecological processes, as these very general descriptors do not provide good explanations to ecological processes, anyway. More detailed work on ecosystem function (beyond empirical relationships) and properties would be necessary to more explicitly link ecosystem function (productivity) to abiotic factors that determine it.

For instance, no details on the selection criteria for climatic variables used in the analysis are presented; no consideration on water limitations (even if they are seasonal, and that determine ecosystem productivity to a great extent) on ecosystems is presented; similarly, MAT may not be the best predictor in a temperate system with highly seasonal precipitation regime; similar questions can be raised for the selection and descriptive metrics of topsoil and surface properties. These are just some examples of questions that be asked about the variables used to disregard the role of climate and surface properties on the dynamics of ecosystem productivity.

The authors propose that geochemistry overwhelms other abiotic factors in explaining ecosystem productivity and that it alone is enough to understand ecological processes, leading to statements related to insensitivity to climate change on carbonate-rich karstic environment. I find this kind of argument unnecessary and, frankly, potentially dangerous for ecosystem management and land-use policy. The geochemical effect that the authors describe relates, ultimately, to water-holding capacity and biogeochemical processes resulting in nutrient availability. Yet, water availability (associated with rainfall variability) is potentially the most pressing uncertainty on climate change effects on ecosystems. To hold water, the regolith needs a water input, which commonly comes from rain (or hydrogeologic fluxes). What happens if this input becomes more scarce or intermittent? if drought becomes more frequent and intense, or if seasonal variability becomes more pronounced, as predicted by most environmental changes projections and supported by observations, ecosystem productivity may be altered. I think the authors don't need to engage in that discussion and present a dichotomy

between climate and other abiotic factors. I find more useful to present their results in the shape of complementary knowledge that deepens our understanding of ecological processes in the face of climate and environmental change.

The paper would be much improved if the methods section was more explicit and detailed. On its current form, multiple aspects of the methodology (particularly those associated with the processes included in the statistical analyses) are rather vague and not straightforward. More specifically, the methods section concentrates on the statistical procedures but leaves multiple questionings about the processes that is trying to relate of describe. In addition, figure legends, both in the main text as well as in the supplementary information could be more effectively used to go beyond a mere description of the graphical elements to make a better description of the main message. The choice of visual and tabular elements in a manuscript of this kind is fundamental for conveying a message as bold as the one the authors intend to send. Perhaps a re-evaluation of the selection of elements would greatly benefit the manuscript.

Point-by-point responses to the comments

Note: text in black are the comments, and text in deep blue are our answers.

We appreciate your constructive comments on our manuscript. We carefully considered each comment and revised the manuscript accordingly. We hope that you will find the revision satisfactory.

Responses to the Reviewer's comments:

TO REVIEWER 1:

1. Title: the statement that geochemistry impacts exceed climate effects is quite strong and, in my opinion, not fully supported by the results.

Answer: We sincerely appreciate the valuable comments. We changed the title to “Bedrock geochemistry influences vegetation growth by regulating the regolith water holding capacity”.

2. Line 23: Earth with capital letter.

Answer: We rewrote this part of the abstract, see line 23.

3. Line 29: Should “Karst” be capitalized?

Answer: This sentence has been corrected, see line 27.

4. Line 34: Write “explained”.

Answer: We rewrote this part of the abstract, see lines 27-38.

5. Line 38: Sounds better to write "...; for 16 of these regions, temperature showed...".

Answer: We rewrote this part of the abstract, see lines 27-38.

6. Line 122-124. Although the RVI for RWLR is indeed higher than for MAT, in explaining variation in NPP, it is only very slightly so (and not quantitatively assessed), why the statement that RWLR is more important than MAT is not fully supported (in my opinion).

Answer: We sincerely appreciate this valuable comment. We realized that the information theoretic approach (ITA) we carried out had some shortcomings since we selected only 2 climatic variables; thus, the importance of the RWLR might not be properly described and exhibit only slightly more plausible importance than the other variables. Hence, considering this comment, we re-built the model subset according to the RVI, it turns out that the RWLR is much higher than that of the MAT (see lines 118-123, 375-396 and Tables 1, 2, S1 and S2).

7. Line 148: But in Figure 4, the correlations between RWLR and NPP are depicted as positive (i.e. no negative sign for correlation coefficients and blue-coloured arrows). What is correct?

Answer: We apologize for our lack of clarity. Actually in the revised manuscript we have clarified that the relationship between the RWLR and NPP is negative because higher a RWLR leads to more and worse drought events during dry spells, which might influence vegetation growth, as described in the revised manuscript. Consequently, the RWLR-NPP relationship has been corrected. Please see Figure 4.

8. Line 157-159: Better to write "Out of the 17 karst zones, 10 showed a significant correlation...", but also give the direction of the correlations. Further, looking at Figure 5, I am not entirely convinced that this is a good argument for the overall conclusion. There will always be variation around a mean, and a few of the 17 sites are quite close to (and below) the global average and 1 site is above. Hence, without any further quantification of the strength of these results (I mean, it is just visually assessed), I question their support for the

strong conclusions taken by the authors (e.g. Line 214-224).

Answer: Thank you for your suggestion. In the revised manuscript, we quantified the difference between karst and non-karst areas by employing a buffer analysis: first, we calculated the NDVI-temperature relationship (R_{NDVI-T}) at the pixel level; second, we defined the buffer zone of each karst area; and finally, we compared the differences in R_{NDVI-T} between karst areas and their buffer zones. We reselected the karst area because (1) karst area numbers 7, 10 and 15 are located in desert, and it is meaningless to assess the importance of bedrock in vegetation productivity in desert areas; (2) the karst area numbers 1, 6, 14 and 16 are too small (less than 30 pixels), resulting in too small sample size of R_{NDVI-T} in order to conduct the proposed robust statistical approach. Therefore, in the revised manuscript, karst area numbers 1, 6, 7, 10, 14, 15 and 16 were removed.

We also rewrote the conclusion and added three paragraphs to discuss the results of the global analysis. For more detail, please see lines 150-160, 210-236, 238-249 and 420-441.

9. Figure 4. Maybe a lighter blue would be better, as the current colour looks close to black. See also comment above about directions of relationships (Line 148).

Answer: Thank you for the suggestion. The colour of Figure 4 has been corrected.

TO REVIEWER 2:

1. First, there is no rationale given for the sites they chose. Why those sites? Is there a study design that these sites satisfy?

Answer: Thank you for your suggestion. The selection of sites was based on the quality of precipitation data. To estimate the regolith water holding capacity of each site, it was necessary to investigate the precipitation patterns and estimate the variation in soil moisture during dry spells, so each CZU should have a meteorological observation station at the centre. There are 71 meteorological observation stations in Guizhou Province, but only 30 of them have precipitation data covering a time span of 10 years (i.e. 2001-2010). We also excluded the sites on which the associated meteorological observation was relocated during 2001-2010 as well as sites characterized by less than 10 identifiable dry spells, resulting in a final selection of 23 sites in total, which satisfy our study design: the selected sites were distributed evenly throughout Guizhou Province, including different bedrock types and along a wide climate range; the sample size was appropriate in order to carry out all the presented statistical analyses because we treated each site as a sample, which makes our statistical approaches robust, and hence, our results are reliable. We added a paragraph to explain the selection of CZU, see line 263-281.

2. Bigger problem: After reading the manuscript, the methods, and the supplemental file and then studying all of the figures and tables as carefully as possible, I still do not know how the authors measured any of their key variables. NPP? RWLR? BrSi? BrCa?

Answer: Thank you for pointing out the weakness of our writing. In the revised manuscript, we added 9 paragraphs to explain how we measured these variables and why we chose them; please see lines 290-374.

3. There is a sense that NPP was maybe measured using remotely sensed NDVI. But how was that done? And there is a sense that RWLR is measured using remotely sensed TDVI. But how exactly? And since these remotely sensed indices are themselves calculated from

different combinations of Landsat bands, doesn't that make it inevitable that they will be strongly correlated (because the same variables are used to calculate both the response and driving factors in the relationship)? And if that's the case, since the NPP-RWLR connection is a linchpin of the paper, doesn't that pretty much call the entire analysis into question.

Answer: Thank you for your comments. We have provided extensive clarification about how we calculated the RWLR in the revised manuscript and supplemented additional data to explain why the NPP-RWLR relationship is not an inevitable correlation.

- (1) In this study, NPP was used to represent vegetation productivity, while NDVI was used to assess soil moisture. Both indices are related to vegetation activity, but they are very different. First, NDVI is an indicator that can be used to represent vegetation growth conditions because it is related to photosynthesis (Tucker, 1979). Even at the same site, the NDVI can change with time; for example, in Guizhou Province, the mean NDVI in July is different from that in December. The NPP used in this study was the annual cumulative value, which therefore is representative for the carbon input throughout the year, which changes very little on a decade scale. Moreover, the formulas for NDVI and NPP are different, because the calculation of NDVI is based on the reflection of the near-infrared and red bands, while the calculation of NPP is much more complicated as it considers not only vegetation activity (7 parameters used for describing vegetation growth) but also climatic factors (Running, et al. 2015).
- (2) We used the TVDI to assess the soil moisture condition, as we estimated the RWLR based on how the TVDI changed during a dry spell (no precipitation for more than 5 consecutive days). The calculation of TVDI is based on the triangle space of temperature/NDVI (Sandholt et al. 2002); the NDVI was used to estimate the maximum temperature under different vegetation conditions. In this circumstance, the correlation between NDVI and TVDI is not inevitable. For example, Schirmbeck et al. (2017) showed that the relationships between NDVI and TVDI are very weak and nonsignificant in bare soil, soybean cropland, grassland, rice paddy, and gallery forest areas. More importantly, the occurrence times of dry spells varied among the CZUs, so the variation in the TVDI during a dry spell also varied among the CZUs, which further decreased the possibility of an inevitable relationship between NDVI and TVDI.
- (3) In summary, given the differences between NDVI and NPP as well as the calculation methods for TVDI and RWLR, we are confident that the relationship between NPP and RWLR is not an artificial correlation. To further test our assumption, we analysed the relationship between NPP and RWLR in Zhejiang Province, which has similar climatic conditions as Guizhou

Province but almost no karst areas (Table 1). The results show that there is no significant NPP-RWLR connection in Zhejiang Province. For more detail, please see Figure 1 below.

Table 1 Comparison of climate and bedrock conditions between Guizhou and Zhejiang

	Zhejiang	Guizhou
Mean annual temperature ()	17.3	15.7
Annual precipitation (mm)	1480	1200
Main bedrock type	clastic rock	carbonate rock

Figure 1 (a) Relationship between NPP and RWLR in Zhejiang Province, which have similar temperature and precipitation condition but belong to non-karst area. The analyzing methods are the same with our manuscript. One plot represents a site with a weather station in its center and 20 km radius. (b) Locations of the two selected provinces Zhejiang and Guizhou.

4. With the BrSi and BrCa, again, I am at a loss for how these were measured. There is difficult to understand conceptual diagram in the supplemental file (Figure S4) that seems like it may provide hints. But I just do not know. Were Si and Ca ratios actually measured somewhere?

Answer: We have added detailed information about the measurement and collection of Si and Ca ratios in the revised manuscript, and we added more detail to the legend of Figure S4; please see lines 299-324 and Figure S4.

5. Beyond the measurements themselves, which are not even reported anywhere in the paper or supplemental files that I had access to (making it impossible to reproduce the analysis), there is the data analysis, which is likewise difficult to follow. I go into some detail about that below, when I address the appropriateness of the statistical measures. Bottom line is, there is no rationale provided for the choices the authors made about how they chose to look for dependencies and codependences in the data. To be honest, my sense after digesting the paper is that they indiscriminately applied a bunch of R modules that sounded to them like they might work and then just treated the individual analyses as black boxes, without really considering whether they were appropriate or not.

Answer: Thank you for your suggestion. We intend to make the storyline more clearly in the revised manuscript. In this study, each individual analysis was carefully discussed. We hypothesized that bedrock chemical composition can influence vegetation productivity through controlling the regolith water holding capacity and tried to describe this lithological control in karst area in three steps. First, to exhibit how important the RWLR is for predicting NPP, we compared the explanatory power with that of other predictor variables. We chose those variables because they are widely considered the most important drivers of vegetation growth. Second, we built two SEMs to show the links among bedrock chemical composition, RWLR and NPP, and we assessed the indirect effect of bedrock composition on NPP. Third, we show the implications of lithological influence for predicting vegetation dynamics under climate change. The influence of bedrock might have the potential to obscure or distort the influence of climate change. This phenomenon is universal and worth further study.

In the revised manuscript, we added all the details about how and why we measured and analysed the candidate variables. In addition, we also assessed the dependencies and codependences in the data carefully (see Table S1). Moreover, we reconsidered the analysis methods and removed some of them in order to make the manuscript more rigorous and concise. See lines 375-377, 397-399, and 420-422.

6. In addition to my concerns about the lack of a clear explanation of how things are measured, a lack of data reporting, and a lack of rationale for why they were analyzed in the way they were, I am surprised that the authors did not spend any time at all discussing what would seem to be an 800-lb gorilla in the room on this question: land use. It seems like it would be a major confounding factor in the analysis of vegetation patterns, especially in China, where deforestation and conversion of landscapes to agriculture have completely altered the ecosystem. This is also likely a confounding factor in the global analysis, but there are even bigger problems with that in my mind.

Answer: Your comments inspired us, and we revised the manuscript accordingly. We analysed the relationship between farmland area and NPP, but we did not find a significant relationship (see Figure 2 at the end of the response letter). In fact, our study area, Guizhou Province, is not an agriculture province. The forest coverage in this area was 57% over the last decade, and this number is still growing; therefore, we believe that the conversion of landscapes to agriculture land use might not be one of primary causes of the spatial variation in vegetation productivity in Guizhou. However, as you mentioned, land use could be a major confounding factor in this study, and it is worthy of a deep discussion. In the karst area, the ecosystem is extremely vulnerable to human activity because of the thin regolith and high permeability of bedrock (Wang et al. 2004), and the landscape is prone to rocky desertification. Therefore, even with the same level of human disturbance, the landscapes in the karst area could be much more easily changed. Consequently, we added a paragraph in the Discussion to explain the influence of the human activity and fragility of the karst ecosystem. See lines 221-228 and Figure 2 below.

Figure 2 Relationship between NPP and percentage of farmland area. The farmland area was obtained from the government annual report of Guizhou Province.

- Regarding the global analysis, the chain of inference that the authors expect us to follow here is: 1) the correlation coefficients in 16 out of 17 karst terrains they analyzed are lower than the global average for correlation between NDVI and mean annual temperature, 2) this means that NPP is controlled by lithology in these terrains. It just does not hold up. One does not follow from the other. There is the statistical issue of how much lower do the coefficients need to be? They do not look very different to me (Figure 5). If they were reported with their uncertainties, is there overlap? We do not know because there is not a proper uncertainty analysis on these correlation coefficients.

Answer: Thank you for your suggestion. In the revised manuscript, we rewrote the explanation about the global analysis: the fact that karst areas are showing weaker NDVI-MAT correlation does not mean that the vegetation production is controlled by lithology; however, this result

suggests that the influence of bedrock might have the potential to obscure or distort the influence of climate change. This seems to be very common, and hence, worth to further study. Moreover, we reanalysed the global dataset using an improved method. I hope you will find the revision satisfactory; see lines 150-160, 210-236 and 420-441 for more details.

8. In summary, from top to bottom — across all the analyses presented here I am unconvinced. The authors had a good idea, but I am not sure what they did to test it. If I follow them correctly, then much of what they did may be wrong, with artifactual correlations (NPP vs. RWLR) and potentially major confounding factors (land use) unaccounted for and not discussed at all in the text.

Answer: Thank you for your critical comments. We explained in one of our answers above why the NPP-RWLR is not an artificial correlation and added a paragraph in order to extend the discussion concerning the influence of land use. Please refer to our answers to your questions 1-7.

9. Overall, the writing in this manuscript is mostly OK, but needs a careful editorial pass to remove some problems like this one on line 230: "...Guizhou Province's lithology, which contains a variety of different ****carbonated**** rock types."

Answer: Many thanks. We have revised the whole manuscript carefully and tried our ultimate best to improve the text structure. In addition, we have asked professional English editors from Springer Nature Author Services to check the English. We believe that the language is now acceptable for the review process.

10. The methods section is grossly inadequate. There is virtually no explanation here of how ANY of the main variables were measured. For example, the method for measuring NPP — one of the two key response variables in the study — is never explained for their study sites in China. For the global analysis, it looks like maybe NDVI is used as a proxy for vegetation. Here is the text on that: "The temperature variation signal of vegetation productivity is calculated as the correlation coefficient between NDVI (obtained from Global Inventory Modelling and Mapping Studies...)." Is this how NPP was estimated at the China sites? We

do not know. We do not even see a data table containing the NPP values. (At bare minimum there needs to be a data table with all of the main variables used in the statistical analyses somewhere in the supplemental file if not in the main text.) Because they do not say and they do not report the basic data needed to understand the work. And if NPP is estimated from the global NDVI data, is that even appropriate? Is it at an appropriate scale?

Answer: Thanks for your suggestion, the NPP dataset was acquired from the Resource and Environment Data Cloud Platform (<http://www.resdc.cn/Default.aspx>), which is a regional scale (whole China) dataset, and NPP was estimated from applying a light energy utilization model. To test the reliability of this dataset, we checked the relationship between the NPP dataset and ground-based data from a recent forest inventory data including 5500 sampling plots. The results show a strong correlation. For more detail, please see lines 284-289 and refer to Figure 3 below.

Figure 3 Relationship between model-based NPP and forest inventory-based NPP was tested at the municipal level, and each plot represents a prefecture or municipal region in Guizhou Province.

11. An explanation of how they quantified the other key response variable is also not given. The authors introduce this “regolith water loss rate” thing out of nowhere as the thing that is controlled by lithology and that in turn controls vegetation, but they never say how it is measured. Nor are values for RLWR reported anywhere. The only reference I can find to how it is measured is the quote: “ RLWR is estimated by evaluating the temporal variation in regolith moisture during dry spell events at the regional scale.” How is regolith moisture measured? From TDVI? But isn’t that just a combination of Landsat bands? And if that is the case, is it not inevitable that there will be a strong correlation between this so called RWLR and NPP, since one is measured from TDVI and the other is presumably measured from NDVI?

Answer: Thank you for your critical comment. The calculation of the RWLR is based on the variation in soil moisture during a dry spell; in this study, the TVDI was used to represent soil moisture. We believe that this remote sensing method is suitable in the context of the present study for the following reasons: first, as the regolith of karst areas is characterized by high spatial heterogeneity and soil moisture monitoring using microclimatic sensors is inappropriate for this purpose because they reflect only the fine-scale environmental conditions; however, remote sensing-based drought indices can provide temporally continuous monitoring and a spatially synoptic view of the soil moisture conditions. Second, TVDI-estimated soil moisture is based on the spatial relationship between land surface temperature and vegetation cover (Sandholt et al, 2002), as the drought in karst areas is foremost the result from high permeability of bedrock and thin regolith, not rainfall scarcity. Hence, the TVDI has been found to be a very appropriate measure for large-scale soil moisture monitoring in karst areas (Zhou et al., 2019); third, and most importantly, the TVDI has been tested in Guizhou, and satisfactory results in terms of soil moisture estimation have been shown (Kang et al, 2008). Please note that the relationship between TVDI and NPP is discussed in further detail in our answers to your question 3. Please also refer to lines 325-396 in the main text and Figure S5 in the Supporting Information.

12. There is also no explanation of how they measured Si or Ca or Mg or Al or Fe concentrations. Are there actual measurements of bedrock geochemistry somewhere? If so why are they not reported? It says soil N is the average of “N-values from the three sampling points.” So the

authors collected 3 soil samples from each CZU? Why is the sampling protocol not described somewhere? Maybe they also sampled bedrock? What about the N analysis procedure? How was this done? I am totally lost as to what the authors did.

Answer: We apologize for our carelessness. The details on the measurement and collection of bedrock and soil element ratios are given in the revised manuscript; please see lines 299-324 and Figure S4.

13. First, the lack of detail provided in the methods summary was not overcome here in the supplementary information. If the authors have shortchanged the methods section in the main text because of length constraints or some other reason, it should be corrected here. But there is no additional text. Just two new poorly explained tables and five new poorly explained figures. And worst of all, they raise more questions than they answer. The figure captions are far too short, failing in all cases to adequately explain the contents of the figure. This is especially true for the concept figures S4 and S5. For example, in figure S4, no information is given about how the supposed lithologic boundaries in the CZUs were identified. Also, precisely what was sampled at the triangles? Bedrock? Soil? Remotely sensed NDVI? The tables also have far too little explanation. For example, it is not clear from table S2 which model is being used. GLM? Mann Whitney Wilcoxon? SEM?

Answer: We apologize for our writing in the previous version. The results of the information theoretic approach are given in Table S2. We also added details about the figures and tables in both the main text and Supporting Information. In particular, the captions of Figures S4 and S5 in the Supporting Information have been greatly improved.

14. It seems that one solution would be to write a brief section about each figure and table. What do the results in each of these pieces show? What part of the paper do they support? Why did you use this test versus another? What are the pitfalls if any of these techniques?

Answer: Thank you for your suggestion. In the revised manuscript, we added several sentences to explain the purpose at the beginning of each statistical analysis.

We hypothesize that bedrock geochemistry can affect vegetation productivity by controlling the regolith water holding capacity. We use Figure 1 to describe our hypothesis in this study. Figure 2 shows the locations of the study area and CZUs.

To test our hypothesis, we first showed how important bedrock geochemistry is in predicting the RWLR and how important the RWLR is in predicting NPP. We compared the explanatory power of the RWLR with that of other climate and soil factors by applying an information theoretic approach (ITA). The results are shown in Table 1, Table S1 and Table S2, highlighting that bedrock geochemistry is the most powerful predictor of the RWLR and that RWLR is the most powerful predictor of NPP. We believe this approach is appropriate because ITA is a model averaging method, and weighted averages of these models can reduce prediction error and better reflect model selection uncertainty. Otherwise, ITA also considers the interaction among candidate variables, which is better than traditional hypothesis testing by a single null model where support is measured by an arbitrary probability threshold. For more details about this approach, see Burnham and Anderson (2002). Figure 3 shows the SEM results. In fact, two SEMs were built based on our hypothesis. The associated results show that BrCa and BrSi have indirect effects on NPP through influencing the RWLR. Hence, as their explanation power for NPP is better than that of climatic and soil factors, these results support our hypothesis that bedrock can strongly influence vegetation productivity in the karst area. The SEM approach has the advantage of estimating the direct, indirect and total effects (direct plus indirect) of explanatory variables on the response variables, which makes this method perfect for testing and describing our hypothesis. For a clearer description of the bedrock-RWLR-NPP relationship, we also analysed the data using a set of independent hypothesis-specific linear models; moreover, the results of single linear models support our SEMs. In Figure 5, we compared the NDVI-MAT relationship between karst and buffer areas using the Mann-Whitney-Wilcoxon test because the data were skewed. The results show that the influence of bedrock might have the potential to obscure or distort the influence of climate change; this phenomenon is universal and is worthy of further study.

In the Supporting Information, Table S1 shows the information on why we selected the MAT, RWLR, Br_{Ca}, Br_{Si} as explanatory variables in the present SEM because they showed a significant relationship with NPP or RWLR. Table S2 shows the best fitting model subsets for NPP and RWLR to support our hypothesis. Figure S1 describes the occurrence period and frequency of dry spells in Guizhou Province. Figure S2 describes the climatic, soil and bedrock conditions in karst and non-karst areas. Figure S3 describes the trend of the TVDI with the length of the dry spell. Figures S4 and S5 are supplementary notes for our sampling and analysis methods. Please see lines 374-377, 397-399, 420-422, for further details.

15. Where to start? Perhaps with the summary: The statistical analyses were largely inappropriate and unreasoned. A snowstorm of techniques was used, but there is not a clear rationale given for any of them. Why is GLM more appropriate than a more straightforward multiple linear regression model? What can we learn from these models? Isn't SEM used with social sciences datasets? Why is it useful here? What are the limitations of these approaches and why are they well suited to analyzing the data presented here (or rather not presented here, since there is no data table anywhere)? Is it appropriate to test every variable you have as a potential explanatory variable? What about the well know problem of multiple comparisons, where the overall false positive rate goes way up? By the way— because the data are not presented anywhere, it would be impossible for anyone to reproduce this work.

Answer: Your suggestions inspired us, and we revised the manuscript accordingly. In the revised manuscript, we changed GLM to a linear regression model. We rewrote the hypothesis and more clearly and concisely described it. Moreover, we added all the raw data in the revised manuscript, and I hope you will find the revision satisfactory.

By comparing those variables with the RWLR, we showed that the RWLR exhibited the best explanatory power for explaining the spatial variation in vegetation productivity. Two SEMs were built based on our hypothesis, which showed that Br_{Ca} and Br_{Si} have indirect effects on NPP by influencing the RWLR. Their explanatory powers for NPP were better than those for climatic and soil factors. In that case, these results support our hypothesis, suggesting that bedrock can strongly influence vegetation productivity in the karst area. The SEM approach has the advantage of estimating the direct, indirect and total effects of explanatory variables on the response variables and has been widely used in ecological studies, which is ideal to test and quantify our hypothesis.

The selection of competing predictor variables was based on previous studies, in which these variables have been widely considered as the main drivers of vegetation productivity in Guizhou Province (Bai et al., 2013; Wang et al., 2008; Zhang et al., 2006). Each variable was carefully re-evaluated to avoid the problem of multiple comparisons. For more detail, please refer to lines 290-294, and Table 1.

16. I am also concerned about the analysis of the global data base of karst terrain. It seems like it is reckless to espouse the chain of inference that the authors seem to want us to follow here. The lack of correlation between NDVI and T means lithology controls NPP in other

landscapes? It's not a chain of inference I would be willing to put any stock in. Moreover, it is not clear that the correlation coefficients are even all that different from the "global average." An uncertainty analysis is warranted here.

Answer: Thank you for your suggestion. To make our results more robust, we thoroughly revised the global analysis by employing a buffer analysis to assess the generality of bedrock control worldwide: first, we calculated the NDVI-temperature relationship ($R_{\text{NDVI-T}}$) at the pixel level; second, we defined the buffer zone of each karst area; and finally, we compared the differences in $R_{\text{NDVI-T}}$ between karst areas and their buffer zones. We reselected the karst area because: 1) karst areas originally coded 7, 10 and 15 are located in the desert, and it is meaningless to assess the importance of bedrock to vegetation productivity in desert areas; and 2) the areas originally coded 1, 6, 14 and 16 are too small to ensure a sufficient sample size for statistical significance, as they were smaller than 30 pixels. Therefore, in the revised manuscript, karst areas originally coded 1, 6, 7, 10, 14, 15 and 16 have been removed. We also updated the discussion on "the lack of correlation between NDVI and T means lithology controls NPP in other landscapes", and changed to show the role of lithology on changing the climatic impact on NPP in the karst areas based on the re-analysis in the revision. For more details, please refer to lines 150-160, 210-236, and 420-441.

TO REVIEWER 3:

1. I find that to be able to solidly ground their proposed hypothesis, the authors need to be more thorough in the use of ecological theory, and provide a better theoretical explanation to their hypothesis. The authors jump very quickly to a general hypothesis from limited ecological data that relates to average climatic conditions and soil information. The authors extend their analysis to a global extrapolation, but the same problems remain in their global consideration. In synthesis, I find that the use of average conditions, as well as only a limited number of climatic and pedological descriptors precludes general descriptions of ecological processes, as these very general descriptors do not provide good explanations to ecological processes, anyway. More detailed work on ecosystem function (beyond empirical relationships) and properties would be necessary to more explicitly link ecosystem function (productivity) to abiotic factors that determine it.

For instance, no details on the selection criteria for climatic variables used in the analysis are presented; no consideration on water limitations (even if they are seasonal, and that determine ecosystem productivity to a great extent) on ecosystems is presented; similarly, MAT may not be the best predictor in a temperate system with highly seasonal precipitation regime; similar questions can be raised for the selection and descriptive metrics of topsoil and surface properties. These are just some examples of questions that be asked about the variables used to disregard the role of climate and surface properties on the dynamics of ecosystem productivity.

Answer: Your comments inspired us, and we revised the manuscript accordingly. We have thoroughly revised our manuscript by considering the ecological theory and provided a better theoretical explanation for our hypothesis. In the revised manuscript, we improved our hypothesis by providing a more reasonable and clearer theoretical explanation. Our main purpose is to emphasize that bedrock geochemistry can be an important regulator of vegetation productivity, but ecologists have tended to give greater emphasis to climate and topsoil factors in the past. Therefore, we selected the climate and topsoil variables that are widely recognized as important factors for vegetation growth and then compared their importance statistically. Taking this opportunity, we highlight that it is indeed necessary to broaden our focus beyond climate and surface soil conditions.

Moreover, we explained the selection criteria for the candidate variables, and all the candidate variables have been considered as the potential drivers of vegetation growth in Guizhou based on previous studies. In addition, we also integrated the Palmer Drought Severity Index (representing water limitations) in the analysis (see lines 290-298, Tables 1 and 2).

2. The authors propose that geochemistry overwhelms other abiotic factors in explaining ecosystem productivity and that it alone is enough to understand ecological processes, leading to statements related to insensitivity to climate change on carbonate-rich karstic environment. I find this kind of argument unnecessary and, frankly, potentially dangerous for ecosystem management and land-use policy. The geochemical effect that the authors describe relates, ultimately, to water-holding capacity and biogeochemical processes resulting in nutrient availability. Yet, water availability (associated with rainfall variability) is potentially the most pressing uncertainty on climate change effects on ecosystems. To hold water, the regolith needs a water input, which commonly comes from rain (or hydrogeologic fluxes). What happens if this input becomes more scarce or intermittent? if drought becomes more frequent and intense, or if seasonal variability becomes more pronounced, as predicted by most environmental changes projections and supported by observations, ecosystem productivity may be altered. I think the authors don't need to engage in that discussion and present a dichotomy between climate and other abiotic factors. I find more useful to present their results in the shape of complementary knowledge that deepens our understanding of ecological processes in the face of climate and environmental change.

Answer: Many thanks. We revised the manuscript accordingly. We rephrased our arguments and added a paragraph to discuss the implications of our result in order to improve our understanding of the observed vegetation dynamics under climate change, see lines 210-236.

3. The paper would be much improved if the methods section was more explicit and detailed. On its current form, multiple aspects of the methodology (particularly those associated with the processes included in the statistical analyses) are rather vague and not straightforward. More specifically, the methods section concentrates on the statistical procedures but leaves multiple questionings about the processes that is trying to relate of describe. In addition, figure legends, both in the main text as well as in the supplementary information could be more effectively used to go beyond a mere description of the graphical elements to make a

better description of the main message. The choice of visual and tabular elements in a manuscript of this kind is fundamental for conveying a message as bold as the one the authors intend to send. Perhaps a re-evaluation of the selection of elements would greatly benefit the manuscript.

Answer: Thank you for your suggestion. In the revised manuscript, we added detailed information about the methods, and improved the figure legends in both the main text and supplementary information, following your suggestions. We also re-evaluated all the predictor variables. I hope you will find the revision satisfactory. Please see lines 290-298, 375-441 and newly added Table 1 in the main text.

REVIEWERS' COMMENTS:

Reviewer #1 (Remarks to the Author):

Dear authors and Editors,

I have now read the responses by the authors to my comment on the original version of the manuscript. I also checked some of their responses to other reviewers' comments. In all, I think the authors have done a good job revising, and thus improving, the manuscript, and, hence, have no further enquiries or comments.

Reviewer #4 (Remarks to the Author):

I have carefully reviewed the revised version of the manuscript, as well as the responses to all comments from the editor as well as the other reviewers. I see that the authors have made a significant effort to address all comments and toned down the paper to reflect what their results indicate. They have improved methods description and have performed better analytical results. However, I still find that their conclusions are based on empirical and statistical analysis over a limited amount of data. Moreover, I still feel the need for mechanistic explanations to their statistical observations, especially when making a bold conclusion with significant ecological and environmental implications. This is particularly relevant when publishing in a prestigious journal like this. In synthesis, although significantly improved, I still think that the manuscript is lacking a more mechanistic exploration of observations. This would increase the potential utility of the results in both modeling and management scenarios.

Point- by- point responses to the comments

Note: texts in black are the comments, and texts in deep blue are our answers.

We appreciate your constructive comments on our manuscript. We carefully considered each comment and revised the manuscript accordingly. We hope that you will find the revision satisfactory.

Responses to the Reviewer's comments:

I have carefully reviewed the revised version of the manuscript, as well as the responses to all comments from the editor as well as the other reviewers. I see that the authors have made a significant effort to address all comments and toned down the paper to reflect what their results indicate. They have improved methods description and have performed better analytical results. However, I still find that their conclusions are based on empirical and statistical analysis over a limited amount of data. Moreover, I still feel the need for mechanistic explanations to their statistical observations, especially when making a bold conclusion with significant ecological and environmental implications. This is particularly relevant when publishing in a prestigious journal like this. In synthesis, although significantly improved, I still think that the manuscript is lacking a more mechanistic exploration of observations. This would increase the potential utility of the results in both modeling and management scenarios.

Answer: We sincerely appreciate your valuable comments. In the revised manuscript, we have strengthened the mechanistic explanation in Discussion. We have carefully strengthened the expression about possible lithological control on vegetation in the Discussion section. Moreover, we have added a separated paragraph to outline the caveats of our study (Line 224-235). We intend to show that bedrock geochemistry can influence the aboveground vegetation at regional scales, which might benefit fine-scale observations in the future. We hope you will find the revision satisfactory.